# Early Lifestyle Interventions in People with Impaired Glucose Tolerance in Northern Colombia: The DEMOJUAN Project

**DOI:** 10.3390/ijerph16081403

**Published:** 2019-04-18

**Authors:** Noël C. Barengo, Tania Acosta, Astrid Arrieta, Carlos Ricaurte, Dins Smits, Karen Florez, Jaakko O. Tuomilehto

**Affiliations:** 1Department of Medical and Population Health Research, Herbert Wertheim College of Medicine, Florida International University, Miami, FL 33178, USA; 2Department of Public Health, Faculty of Medicine, University of Helsinki, 00100 Helsinki, Finland; Jaakko.tuomilehto@helsinki.fi; 3Department of Public Health, Universidad del Norte, Barranquilla 080001, Colombia; tacosta@uninorte.edu.co; 4Centro de Investigation Sanitaria, Barranquilla 080001, Colombia; astridisabel1@gmail.com (A.A.); carlos.ricaurte@gmail.com (C.R.); 5Faculty of Medicine, Riga Stradins University, LV-1007 Riga, Latvia; dins.smits@rsu.lv; 6Departamento de Matemáticas y Estadística, Universidad del Norte, Barranquilla 080001, Colombia; lozanok@uninorte.edu.co; 7Department of Public Health Solutions, National Institute for Health and Welfare, 00271 Helsinki, Finland; 8Saudi Diabetes Research Group, King Abdulaziz University, Jeddah 21589, Saudi Arabia

**Keywords:** glucose metabolism disorders, primary prevention, South America, population, field trial

## Abstract

Background: The objective of the demonstration project for type 2 diabetes prevention in the Barranquilla and Juan Mina (DEMOJUAN) study was to investigate the extent to which it is possible to reach normal glucose metabolism with early lifestyle interventions in people at high risk of type 2 diabetes (prediabetes), compared with those who receive standard usual care. Methods: DEMOJUAN was a randomized controlled trial conducted in Juan Mina and Barranquilla, Northern Colombia. Eligible participants were randomized into one of three groups (control group, initial nutritional intervention, and initial physical activity intervention). The duration of the intervention was 24 months. The main study outcome in the present analysis was reversion to normoglycemia. Relative risks and their corresponding 95% confidence intervals were calculated for reversal to normoglycemia and T2D incidence. Results: There was no statistically significant association between the intervention groups and reversion to normoglycemia. The relative risk of reversion to normoglycemia was 0.88 (95% CI 0.70–1.12) for the initial nutritional intervention group participants and 0.95 (95% CI 0.75–1.20) for the initial physical activity intervention group participants. Conclusions: Our study did not find any statistically significant differences in reversion to normoglycemia or the development of type 2 diabetes between the intervention groups and the control group in this population.

## 1. Introduction

The International Diabetes Federation (IDF) has estimated that the number of adults with diabetes is expected to rise from 425 million in 2017 to 629 million by 2045 [1]. According to recent data in 2017, the national prevalence of type 2 diabetes (T2D) in Colombia was 8.1% for the 20- to 79-year-old population, imposing a heavy financial burden on the primary healthcare system [1]. Premature mortality caused by T2D may lead to 12–14 years of life lost [2].

Randomized controlled trials carried out in different populations have successfully revealed that, in people with impaired glucose tolerance (IGT), the progression to T2D can be prevented or delayed by lifestyle intervention [3,4,5,6,7,8,9]. These studies have revealed that nutritional and physical activity interventions can decrease the relative risk of T2D by 30–60% in people with IGT in controlled settings [3,4,5,6,7,8,9]. The length of the interventions of previously mentioned trials varied between 2.5 and 6 years [3,4,5,6,7,8,9]. However, none of these studies included Latin American populations.

Several studies have investigated whether the results of these randomized clinical trials can be implemented within the primary healthcare system [10,11,12,13,14,15,16,17,18]. However, the main outcome assessed by these translational research projects was weight loss, not conversion to normoglycemia, which should have been the main target. Weight loss was greater, in almost all studies, in the intervention groups than with the controls. In addition, the majority of the studies did not find differences in blood glucose or waist circumference. Scientific evidence regarding T2D prevention is lacking from Latin America.

While previous findings offer a compelling evidence base, it is necessary to learn how the prevention of T2D works in real life in different settings and population groups. In addition, it is important to find out how well the lifestyle interventions work within the primary healthcare system in Latin America. Thus far, such trials have not been carried out in the Caribbean region of Latin America and it is not known whether these lifestyle interventions will work well in a hot and humid climate, particularly in the physical activity component. The extreme weather conditions seen in the Caribbean region (thunder storms and humid conditions) may need special logistics and motivation to improve daily physical activity of the study population.

The main objective of this analysis of the data from the demonstration project for T2D diabetes prevention in the Barranquilla and Juan Mina (DEMOJUAN) study was to investigate the extent to which it is possible to reach normal glucose metabolism with early lifestyle interventions in people at high risk of T2D, compared with those who receive standard therapy (usual care).

## 2. Materials and Methods

### 2.1. Study Design and Oversight

The design of the DEMOJUAN study was a three-arm, parallel-group, randomized, and controlled field trial, using individual random allocation, and was carried out in Juan Mina and Barranquilla, Northern Colombia [19]. The study site had approximately 1 million inhabitants at the time of participant recruitment. A trial coordinating center in Barranquilla served as a data and statistical analysis center and supervised all study sites where patients were recruited and included in the trial.

This project was supported by a BRIDGES Grant from the Global Diabetes Foundation. BRIDGES, an International Diabetes Federation project, is supported by an educational grant from Lilly Diabetes. This study followed the Good Clinical Practice guidelines and the guidelines of the Helsinki Declaration. All data were collected using previously tested questionnaires and methods as much as possible. Aside from blood samples, no invasive methods were used. The study protocol was approved by the research ethics committee of the University Pontificia Javeriana, Bogota, Colombia. All the participants gave their written informed consent prior to participation in the study. The steering committee designed the study, gathered the data (in collaboration with investigators at the local clinics and other study units), made the decision to submit the manuscript for publication, and guaranteed the compliance of the study with the protocol.

The trial was registered at https://clinicaltrials.gov/ (NCT01296100 (2/12/2011)). This work has been developed following the guidelines of the CONSORT statement, the evidence-based minimum set of recommendations for reporting randomized trials [20].

### 2.2. Study Population

The study participants were recruited in the study sites (Juan Mina and Barranquilla) by population screening, using the Finnish Diabetes Risk Score (FINDRISC) [21,22,23]. The FINDRISC is based on easily available information using eight parameters with categorized answers. The total risk score value ranges from 0 to 26. A FINDRISC of >13 points was considered to be a high risk of having IGT/IFG or diabetes. The FINDRISC was originally shown to predict the 10-year risk of drug-treated T2D with a sensitivity of 78–81% and a specificity of 76–77% [21]. Furthermore, it also detects reasonably prevalent asymptomatic T2D and other disorders of glucose metabolism [23]. The FINDRISC has been validated in many populations with good results [24,25,26,27] and has been successfully applied in primary care in Barcelona, Spain [19]. It is recommended by the International Diabetes Federation and in the guideline by the European Society for the Study of Diabetes and the European Society of Cardiology as a screening tool for T2D [28,29]. In our study, all people with more than 13 FINDRISC points were invited to an oral glucose tolerance test (OGTT). 

Participants 34–69 years of age with a 2-h post-challenge glucose level of 140–200 mg/dL (IGT) or a fasting plasma glucose level between 100 and 126 mg/dL (impaired fasting glucose) were included in the study. In addition to the initially defined inclusion criteria when registering the trial, we also accepted participants with isolated impaired fasting glucose (*n* = 109) into the study. The reasons for these changes were numerous. Firstly, people with IFG have a similar risk of progressing to diabetes as people with IGT. Secondly, there is a lack of knowledge regarding the prevention potential in people with IFG, since almost all diabetes prevention trials have used IGT as the entry criterion. People with pharmacologically treated diabetes and hypertriglyceridemia while undergoing drug treatment, a history of life-limiting diseases or events, and an unwillingness to sign the informed consent were excluded from the study.

### 2.3. Randomization and Interventions

Eligible participants were randomly allocated to one of three groups (A, B, or C) with 1:1:1 allocation. The sequences for the random allocation groups were generated by IBM SPSS statistics version 19.0 for Windows (IBM Corp.: Armonk, NY, USA). The randomization list was prepared by an independent statistician. Participants and study personnel were aware of the study-group assignments, but outcome adjudicators were not.

During the following 24 months, the control group (A) received standard treatment (usual care: Lifestyle advice prescribed by his/her physician). In addition, the participants in the control group were told about their high risk of T2D and received a “mini-intervention” to improve their lifestyle. The two intervention groups (Groups B and C) received early intensive lifestyle interventions. Group B received first a nutritional intervention for 6 months, followed by the physical activity intervention (6 months), and a combined nutritional and physical activity intervention (12 months). They were also provided with individual advice about how to achieve the intervention goals. These goals were (i) reduction in weight of >5%, (ii) total fat intake less than 30% of energy consumed, (iii) saturated fat intake less than 10 % of energy consumed, and (iv) fruit or vegetable intake of at least 500 g per day. The participants of group C started with the physical activity intervention (6 months) followed by the nutritional intervention (6 months). The physical activity intervention consisted of individual visits with a physical activity specialist and monthly group seminars. Each participant had six individual visits with a physical activity specialist (four times during the first year and twice in the second year), where each participant received an individual physical activity prescription. The goal of the physical activity intervention was to practice moderate-intensity exercise for 30 min a day or more. In addition, the participants of Groups B and C were divided into groups of 10 participants for the group seminars, too. The group seminars were held monthly during the first 12 months of the intervention and then every second month during the second year of the intervention. The group seminars were given by a nutritionist and physical activity specialist. During the lifestyle intervention period, the study participants had two study visits at month 1 and 24, respectively. The OGTT was done at month 1 and month 24. In order to reduce the possibility of bias due to visiting schedule, both intervention groups had their group sessions and individual visits in parallel, at the same time. However, the seminars were held independently for the participants of Groups B and C. The description of the topics of the lifestyle intervention program are presented in Appendix A.

### 2.4. Study Measurements

#### 2.4.1. Non-Invasive Measurements

Lifestyle habits and risk factors for T2D were assessed by an interview, using a questionnaire consisting of information regarding sociodemographic factors, history of T2D, medical history, tobacco consumption, hypertension, and nutritional and physical activity habits. The instruments applied were designed based on FINDRISC, the World Health Organization Stepwise approach to surveillance (STEPS) of non-communicable disease risk factors, and International Physical Activity Questionnaire (IPAQ) [30,31]. All questionnaires have been successfully validated in large international studies [32,33,34,35,36]. Physical activity measured with the questionnaire shows a high correlation with physical fitness, measured by maximal oxygen uptake [34,35,36]. Dietary habits were assessed by 16 questions (e.g., dietary pattern, quality and quantity of dietary fat, consumption of fruit and vegetables, grains, milk and meat products, desserts, sweets, and alcoholic beverages). In addition, 6 questions were related to perceived needs and intentions to make dietary changes. Scientific validation of the diet questionnaire was carried out in the National Institute for Health and Welfare in Helsinki, Finland [37]. The study questionnaire can be found in Appendix A.

All measurements followed a standardized protocol and the staff taking the measurements were professionals with long-standing experience in measuring anthropometry and blood pressure. Height and weight were measured without shoes and with light clothing. BMI was calculated as weight (kg) divided by height^2^ (m^2^). Waist circumference (to the nearest cm) was measured at the approximate midpoint between the lower margin of the last palpable rib and the top of the iliac crest. The categories, <94 cm vs. ≥94 cm in men and <90 cm vs. ≥90 cm in women, were used to identify abdominal obesity in the Latin American population [38]. Blood pressure (2 mm Hg precision) was recorded twice with a mercury sphygmomanometer in a seated position. 

#### 2.4.2. Biochemical Measurements

All participants underwent an OGTT that was carried out according to the World Health Organization (WHO) recommendations [39]. The test started after at least 12 h of fasting, and the fasting and 2-h blood samples were obtained after oral ingestion of water solution with 75 g anhydrous glucose. The glucose tolerance status was classified according to the criteria of the American Diabetes Association [40]. Individuals who had a fasting plasma glucose level ≥126 mg/dL or 2 h plasma glucose (2hPG) level ≥200 mg/dL were classified as having T2D. Those with 2hPG ≥ 140 mg/dL but <200 mg/dL, and FPG < 100 mg/dL were classified as having isolated IGT. Isolated IFG was defined as FPG ≥ 100 but <126 mg/dL, and 2hPG < 140 mg/dL. People with 2hPG ≥ 140 mg/dL but <200 mg/dL, and FPG ≥ 100 but <126 mg/dL were defined as having combined IGT and IFG. People with T2D, IGT, or IFG were classified as having IGT.

### 2.5. Study Outcomes

The main study hypothesis was that the lifestyle interventions would lead to a higher reversion to normoglycemia, defined as FPG < 100 mg/dL and 2hPG < 140 mg/dL, compared with the control group.

We also evaluated the development of T2D in the study participants.

### 2.6. Statistical Analysis

The study was designed to have a 90% power to detect a 20% percentage unit difference in recovery from IGT (comparing 70% vs. 50% recovery rates) between the treatment groups at 5% significance level. This estimate was chosen due to previous scientific evidence revealing that, as compared with the placebo, pharmaceutical intervention might increase the conversion rate to normoglycemia from 50 to 70% in people with IGT [7,41,42,43]. Assuming a 30% loss to follow up at the end of the 24-month intervention, a total of 200 participants were needed in both treatment groups (total sample size of 600 individuals). The estimated drop-out of 30% was decided according to the results and experiences of previous randomized controlled diabetes trials in Europe and the Unites States [6,11,44]. The data was analyzed using SPSS statistics version 19.0 for Windows with use of the intention-to-treat approach for all randomly assigned participants. The variables were checked for normality using Kolmogorov–Smirnoff tests. The χ² test was used to test differences in the distribution between categorized variables. Collinearity diagnostics were used to check for correlations between the variables. Relative risks and their corresponding 95% confidence intervals for the intervention groups compared with the control groups were calculated for reversal to normoglycemia and T2D incidence. Differences in FPG and 2hPG among the three groups were assessed using mixed-design analysis of variance models in order to adjust the estimates for the difference in glucose status at baseline. The results are expressed as percentages, means, and standard errors/standard deviations. The threshold for statistical significance was set to 0.05.

## 3. Results

Figure 1 shows the eligibility, random allocation, and follow-up of the study participants. In total, 14,193 people from the community were screened using the FINDRISC. Those with a FINDRISC of at least 13 points were invited to an OGTT (*n* = 4915) and approximately 50% of those invited attended. Among those with an OGTT result, 1535 did not fulfill the entry criteria of the study, as they had either normoglycemia or screen-detected T2D. A total of 772 participants were randomly allocated to the three groups: (A) Control group (*n* = 246); (B) initial nutritional intervention group (*n* = 261); and (C) initial physical activity intervention group (*n* = 265), respectively. Half of the study participants were lost during the follow-up of at least 18 months, the final number of study participants that entered the data analysis process was 122 for the control group (50%), 136 in the nutritional intervention group (52%), and 132 in the physical activity intervention group (50%).

Table 1 presents the baseline characteristics of the study participants after randomization; these characteristics were mostly equally balanced among the three groups, indicating that the randomization process worked well in controlling possible confounders. 

When assessing the baseline characteristics of the three groups for the study participants included in the analysis (Table 2), it was found that all characteristics were equally distributed, with the exception of glucose metabolism (*p*-value < 0.001) at baseline. More study participants in the physical activity intervention group (36%) had combined IFG and IGT compared with the nutritional intervention group (21%) or the control group (16%). Normoglycemia and isolated IFG tended to be more frequent in the control group compared with the two intervention groups.

Table 3 reveals the unadjusted and adjusted results of the logistic regression analysis for the primary and secondary outcomes of the study. There was no statistically significant difference among the intervention groups in reversion to normoglycemia. The relative risk of reversion to normoglycemia was 0.88 (95% CI 0.70–1.12) for the initial nutritional intervention group participants and 0.95 (95% CI 0.75–1.20) for the initial physical activity intervention group participants. Similarly, no statistically significant association was found among the intervention groups in the incidence of type 2 diabetes. The corresponding relative risk for the nutritional intervention group was 1.38 (95% CI 0.67–2.84) and 1.43 (95% CI 0.70–2.93), respectively.

The participation rates in the intervention sessions varied according to the type of session (Table 4). In general, three out of four intervention group participants took part in at least one group and one individual intervention session. Whereas 32% of the nutritional intervention group participants came to at least 50% of all planned individual sessions, only 9% had at least a 50% participation rate for the group sessions. The corresponding participation rates for the physical activity study participants were 46% (individual sessions) and 15% (group sessions).

Finally, we also calculated the changes in FPG and 2hPG between baseline and after follow-up, according to intervention group (Table 5). No statistically significant changes were observed in fasting glucose levels in the three groups (*p*-value 0.644). However, all study participants showed a decrease in their 2hPG independently of the group they were assigned to initially (overall *p*-value < 0.001). The decrease in 2hPG was 19 mg/dL for the control group, 14 mg/dL for the nutritional intervention group, and 29 mg/dL for the physical activity intervention group. The *p*-value for the interaction (intervention and fasting and 2-h glucose levels between baseline and after follow-up) was 0.201, indicating that no differences in the changes of the glucose levels were observed between the intervention and control groups.

## 4. Discussion

Our study did not find statistically significant differences either in reversion to normoglycemia or the development of T2D between the intervention groups and the control group. Moreover, 2hPG was reduced in all three groups, whereas their FPG remained unchanged between baseline and the end of the study. This is in line with findings from other studies where changes in FPG following lifestyle intervention has been marginal.

Several studies have tried to implement the findings and experiences of randomized controlled trials within the primary health-care system [10,11,12,13,14,15,18]. The DEPLOY study evaluated the delivery of a group-based Diabetes Prevention Program (DPP) lifestyle intervention in partnership with the YMCA [10,11]. After six months, body weight decreased by 6.0% (95% CI = 4.7, 7.3) in intervention participants and 2.0% (95% CI = 0.6, 3.3) [11]. These changes were maintained through 25–32 months of follow-up [10]. The study initially included 92 study participants and their documented loss in follow-up during the first 12 months was 37% for the intervention group and 28% for the control group. A systematic review and meta-analysis of twenty-eight US-based studies applying the findings of the DPP revealed that the average weight change at twelve months after the intervention was a loss of about 4% from participants’ baseline weight [12]. Unlike our study, they did not measure fasting or 2hPG during the follow-up. The Diabetes in Europe: Prevention using Lifestyle, Physical Activity and Nutritional Intervention (DE-PLAN) project aimed at implementing the strategy of the Finnish landmark Diabetes Prevention Study (DPS), within several countries in Europe [13,14,18] in high-risk people identified with the FINDRISC. Similar to our study, those studies had either T2D or reversion to normoglycemia as a main study outcome [13,14,18]. The first large-scale nationwide diabetes prevention program in the world, the National Program for the Prevention of T2D (FIN-D2D), was implemented in Finland during 2003–2008 and evaluated in five hospital districts [18]. They revealed that the relative risk of T2D was 0.31 (95% CI 0.16–0.59) in the group who lost ≥5% weight, 0.72 (0.46–1.13) in the group who lost 2.5–4.9% weight, and 1.10 (0.77–1.58) in the group who gained ≥2.5% compared with the group who maintained weight [18]. The DE-PLAN group in Poland included 175 middle-aged, slightly obese participants in nine primary healthcare centers in Krakow [14]. In keeping with our study results, they did not find changes in FPG or 2hPG in people who participated in the interventions, but, similar to US-based implementation studies, the Polish intervention led to a significant reduction in weight during the first 12 months [14]. A prospective cohort study in the setting of Spanish primary care found that intensive lifestyle interventions were feasible in a primary care setting and substantially reduced T2D incidence among high-risk individuals [13]. During a 4.2-year median follow-up, they revealed a 36.5% relative risk reduction in the incidence of T2D between the intervention and control group. The study in the Greater Green Triangle of Southwest Victoria and Southeast South Australia in 2004–2006 using general practices provided evidence that a T2D prevention program using lifestyle intervention was feasible in primary health care settings, with reductions in risk factors approaching those observed in clinical trials, observing T2D in adults with IGT [15]. Unlike in our study, they reported reductions in FPG and 2hPG in their study participants during their 12-month intervention.

One important factor explaining the differences of the results between randomized controlled clinical trials and their implementation in the general population is length of intervention. Whereas the length of the interventions of the randomized clinical trials varied between 2.5 and 6 years [3,4,5,6,7,8,9], the duration of the interventions of the pragmatic or field trials in the “real world” were of shorter duration [10,11,12,13,14,15,16,17,18]. Implementing randomized controlled clinical trials in the population setting has been shown to have less impact, among others, due to a higher heterogeneity of the study population and compliance with the interventions. Furthermore, compared with randomized controlled trials, it is very difficult to reach a very high compliance (or participation rate) with the interventions in real –life field trials as the study population cannot be controlled in a similar way. However, our study showed that interventions can be implemented within the primary healthcare system and that at least reaching reasonable compliance rates for individual sessions may be achieved. Naturally, the reasons for non-compliance need to be assessed further in order to adapt the program. Thus, this and our short length of intervention may be some additional reasons why we were not able to show any differences in changes of the glucose levels between the intervention and the control group.

The problem with translational research in implementing principles of the proof-of-concept randomized clinical controlled trials in real-life settings is that many factors that can be controlled in trials cannot be done in routine clinical practice. The political agenda to work on diabetes prevention, what we believe triggered our project, affects the entire population. Nevertheless, it was of surprise that an improvement in the glucose profile was observed in both the intervention and control group. Several reasons may explain these findings. Firstly, probably just participating in this study may encourage people in the control group to adopt a healthier lifestyle, as also observed in previous studies (Hawthorne effect). Actually, the participants in our control group were told about their high risk of T2D, and they also received a “mini-intervention” to improve their lifestyle. Moreover, in some instances, members of the three groups were from the same community, thus, a dilution of the intervention from the intervention groups to the control group participants may have occurred. Secondly, during the study, many activities were carried out by the city of Barranquilla targeting people with glucose metabolism disorders. For instance, environmental changes were implemented in the city in the form of creating parks and special sites where people could practice physical activity. In addition, there were steady information campaigns targeting people with overweight and obesity to adopt a healthier lifestyle to avoid T2D. Diabetes as a health topic was omnipresent during the last four years in Barranquilla, until the Ministry of Health of Colombia declared Barranquilla the national demonstration area for diabetes prevention. Yet another potential explanation of the results may be partly a regression to the mean of the 2hPG that cannot be fully excluded. However, as study participants were randomly allocated to the intervention and comparison groups, the impact of the intervention would be equally affected by regression to the mean [45].

One of the strengths of our study was that we managed to design and implement a field trial within a primary healthcare system of a country in economic transition. Close to 50% of the people with prediabetes at the beginning of the study had their glucose values normalized at the end of the study. This shows that early lifestyle intervention programs may prevent or delay the development of T2D and can be successfully integrated into the primary healthcare system. Lifestyle intervention trials conducted in a controlled research setting have successfully shown that the incidence of T2D in high-risk individuals with IGT can be approximately halved (DPS and DPP studies). Transversal implementations of the DPP within YMCA have shown similar results to ours, revealing an improvement especially in the risk factors of T2D, such as overweight/obesity and levels of physical activity [11,12].

As a limitation, we were not able to assess changes in lifestyle or anthropometric measurements in all of the study participants. In addition, the follow-up time of the participants ranged between 18 and 24 months only. Furthermore, the lack of generalizability to populations not included in the study, such as persons from other provinces of Colombia, may limit external validity to the entire Colombian population. Finally, as we lost close to 50% of the study participants during the follow-up, our study may not have sufficient power to show a difference in reversion to normoglycemia or development of T2D.

## 5. Conclusions

In conclusion, the DEMOJUAN project showed some benefits for glycaemia with significant 2hPG fall, but failed to link it with the lifestyle interventions provided, compared with the control group. Nevertheless, it has to be kept in mind that various factors affect the outcome of intervention programs in a real population setting, as the conditions cannot be adequately controlled, such as in clinical controlled trials. Therefore, this study adds an important piece of information to existing knowledge resulting from lifestyle interventions of clinically controlled trials that can successfully be implemented in real-life settings. Future studies should include an OGTT, combined with information on changes of lifestyle and risk factors of T2D, of the study participants even when conducted within the primary healthcare system.

## Figures and Tables

**Figure 1 ijerph-16-01403-f001:**
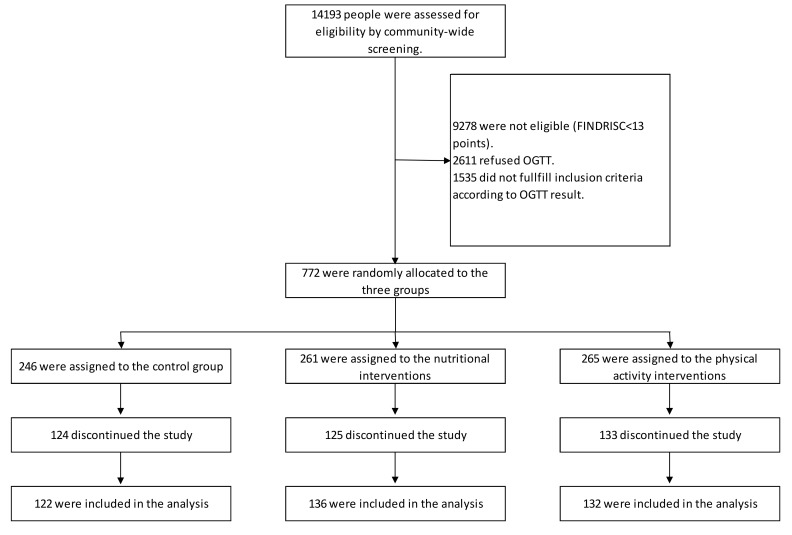
Eligibility, random allocation, and follow-up of the study participants. Discontinued intervention refers to participants who discontinued, due to being lost in the follow-up.

**Table 1 ijerph-16-01403-t001:** Baseline characteristics of the study participants after randomization.

	Intervention Groups			
	Control	Nutrition	Physical Activity	*p*-Values
	(*n* = 246)	(*n* = 261)	(*n* = 265)	
	mean (SD ^1^)	mean (SD)	mean (SD)	
**Age**	53.8 (9.1)	52.9 (9.3)	52.2 (8.1)	0.105
	**% (*n*)**	**% (*n*)**	**% (*n*)**	
**BMI** ^1^				0.218
<25 kg/m^2^	15 (37)	17 (43)	11 (30)	
25–30 kg/m^2^	36 (88)	41 (106)	44 (115)	
>30 kg/m^2^	49 (121)	43 (111)	45 (119)	
**Waist circumference**				0.59
≥94 cm (men)/≥90 cm (women)	89 (219)	88 (229)	86 (228)	
**>30 min physical activity/day**	13 (31)	13 (35)	15 (40)	0.703
**Daily consumption of fruits and vegetables**	23 (57)	20 (51)	23 (61)	0.528
**Anti-hypertension drug use**	46 (114)	44 (115)	46 (121)	0.868
**Family history of diabetes**				0.657
No	29 (71)	23 (61)	25 (67)	
Yes: Grandparent, uncle, aunt, or cousin	22 (54)	26 (67)	23 (61)	
Yes: Biological father, mother, or sibling	49 (121)	51 (132)	52 (136)	
**Previous increased glucose in blood**	28 (69)	30 (77)	35 (93)	0.187
**Glucose metabolism**				
Normoglycemia	18 (22)	11 (15)	13 (17)	0.069
Isolated IFG	32 (40)	28 (57)	23 (38)	
Isolated IGT	35 (44)	41 (38)	29 (30)	
IFG and IGT	15 (19)	20 (28)	35 (47)	

^1^ Standard deviation.

**Table 2 ijerph-16-01403-t002:** Baseline characteristics of the study participants included in the analysis.

Effectively Analyzed	Control	Nutrition	Physical Activity	*p*-Values
	(*n* = 122)	(*n* = 136)	(*n* = 132)	
	**mean (SD ^1^)**	**mean (SD)**	**mean (SD)**	
**Age (years)**	53.7 (8.9)	52.3 (9.1)	52.6 (8.2)	0.420
	**% (*n*)**	**% (*n*)**	**% (*n*)**	
**BMI ^1^**				0.732
<25 kg/m^2^	13 (16)	18 (24)	13 (17)	
25–30 kg/m^2^	43 (52)	40 (55)	46 (61)	
>30 kg/m^2^	44 (54)	42 (57)	41 (54)	
**Waist circumference**				
≥94 cm (men)/≥90 cm (women)	90 (110)	88 (120)	82 (108)	0.118
**>30 min physical activity/day**	12 (15)	12 (16)	17 (22)	0.444
**Daily consumption of fruits and vegetables**	28 (34)	19 (26)	24 (32)	0.249
**Anti-hypertension drug use**	45 (55)	42 (57)	46 (60)	0.815
**Family history of diabetes**				0.618
No	30 (37)	25 (34)	26 (35)	
Yes: Grandparent, uncle, aunt, or cousin	17 (21)	25 (34)	23 (30)	
Yes: Biological father, mother, or sibling	53 (64)	50 (68)	51 (67)	
**Previous increased glucose in blood**	27 (33)	27 (36)	36 (48)	0.145
**Glucose metabolism**				<0.001
Normoglycemia	18 (22)	11 (15)	13 (17)	
Isolated IFG	35 (43)	27 (36)	23 (30)	
Isolated IGT	31 (38)	42 (57)	29 (38)	
IFG and IGT	16 (19)	21 (28)	36 (47)	

^1^ Standard deviation.

**Table 3 ijerph-16-01403-t003:** Primary and secondary study outcomes.

	Control		Nutrition	Physical Activity
	(*n* = 122)		(*n* = 136)	(*n* = 132)
	RR ^1^	(95% CI ^2^)	RR (95% CI)	RR (95% CI)
**Primary outcome**				
Reversion to normoglycemia	1	Ref. ^3^	0.88 (0.70–1.12)	0.95 (0.75–1.20)
**Secondary outcome**				
Type 2 diabetes	1	Ref.	1.38 (0.67–2.84)	1.43 (0.70–2.93)

^1^ Risk ratio; ^2^ Confidence interval; ^3^ Reference group.

**Table 4 ijerph-16-01403-t004:** Participation rates of the study participants in the interventions.

Group	Control	Nutrition	Physical Activity
	(*n* = 122)	(*n* = 136)	(*n* = 132)
	% (*n*)	% (*n*)	% (*n*)
Participated in at least one group and one individual intervention session	0 (0)	74 (100)	76 (100)
Participated in at least 50% of all individual sessions	0 (0)	32 (43)	46 (61)
Participated in at least 50% of all group sessions	0 (0)	9 (12)	15 (20)

**Table 5 ijerph-16-01403-t005:** Changes in fasting and 2-h glucose levels between baseline and after follow-up, according to intervention group.

	Control		Nutrition		Physical Activity	
	(*n* = 122)		(*n* = 136)		(*n* = 132)	
	Mean	SD ^1^	Mean	SD	Mean	SD
**Fasting glucose**						
Baseline	97	(10)	101	(10)	102	(11)
After follow-up	101	(19)	102	(21)	106	(32)
***p*-value for difference between baseline and follow-up = 0.644**						
**2-h glucose**						
Baseline	143	(25)	141	(26)	146	(26)
After follow-up	122	(43)	127	(52)	117	(49)
***p*-value for difference between baseline and follow-up <0.001**						
*p*-value for differences between intervention groups 0.267						
*p*-value for interaction (intervention and fasting and 2-h glucose levels between baseline and after follow-up group) 0.201						

^1^ Standard deviation.

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
