# Peer review of "Early Lifestyle Interventions in People with Impaired Glucose Tolerance in Northern Colombia: The DEMOJUAN Project"

_ijerph, 2019, doi:10.3390/ijerph16081403_

Round 1

Reviewer 1 Report

COMMENTS TO THE PAPER:

This is manuscript trying to present the results of a pragmatic clinical trial assessing early lifestyle interventions to reach normal glucose levels on those with prediabetes. In addition, the potential benefit of the intervention on the risk of developing T2DM was also evaluated.

The main concern of this reviewer is focused on the presentation of some of the results of the trial and not all of them as provided in the clinicaltrials.gov registry. This is important as this registry shows what the authors decided to assess in this trial. In addition, there is not clear information about several of the items of the CONSORT guideline for trials, for example, selection criteria of participants, use of the intent-to-treat principle for analyses, among others.

Major comments:

Abstract:

- Good to clarify the duration of the intervention in some part... was that enough for glycemia reversion?

-  Prefer to use RR instead of OR to report results.

Introduction:

- Line 45-49: Clarify the duration of the mentioned interventions. Clarify the range of potential effect reported in line 48 (I suppose 48% is the maximum effect).

-  Line 54-55: Add references according to the studies reported.

- Line 61-62: Please explain why the interventions are not expected to work well in hot and humid climate?

Materials and methods:

- Please explain what were the inclusion and exclusion criteria for the study? This part needs to coincide with the report in clinicaltrials.gov (if not, please explain why?)

- Give more information about the design of the study (what kind of trial was?) and about randomization (who did the randomization?  Was a third party? Or some from the team?). Who did the assignment?

-  Line 114-116 and line 121-122: some goals are reported there. Were they assessed? Were they considered as possible secondary outcomes? This can help to understand why the intervention did not work...

- Line 123-125: were the participants from arm B and C combined for the seminars? How the seminars were done? What were the topics? What was the duration of the seminars? Some examples can help to understand that. In addition, a figure could help to understand what happen in each arm on the intervention...

- Line 138-140: the paper will benefit if the questionnaire can be added to the manuscript as online supplement or something similar. Although the questionnaires were validated, questions can be relevant to understand why the intervention did not work.

- Line 161-164: Study outcomes: This part needs more information: Was any new OGTT done? When? Every 6 months? Or only at the end? What were the secondary outcomes if any (please check clinicaltrials.gov)? How many times was OGTT measured over time? Only at baseline and at the end? Every six months? These topics need to be clarified!

- Secondary outcome according to clinicaltrials.gov is not considered in the analyses: "Difference in reduction of 10 year-estimated cardiovascular risk score of 10% between the standard therapy control group and the lifestyle intervention groups (WHO CVD prediction chart)". Why not to include it in the manuscript?

- Line 166: How was the 20% difference estimate chosen? Is that estimate relevant?

- Line 167: Is the comparison between groups? Or is it between intervention groups compared to the control one? Was the intent-to-treat principle followed in the analysis?

- Line 175: Why OR are calculated instead of RR? I think the manuscript will benefit to have the correct estimate here. Similarly, why the ANOVA was used instead of mixed models? In this case is required for the need of adjust estimates for difference in glucose status at baseline. Why not the secondary outcomes were assessed according to the clinicaltrials.gov?

Results:

- Relative risk should be the estimate to be reported (instead of OR).

- The interventions should have a positive effect on the outcome, i.e. increase the probability to revert to normoglycemia, and reduce the possibility to develop T2DM... was the codification well done in analyses?

- There is no clear information about the use of repeated measures or the use of intent-to-treat principle.

- Where are the secondary outcomes (please use info of clinicaltrials.gov, all need to be reported).

-  Figure 1: More than 50% of lost to follow-up; at the end, the study was underpowered to find the proposed differences. How affect results?

-  Table 1: what that the best way to present the age? Why not as numerical variable (i.e. mean and SD)?

-  Table 4: Were these estimates adjusted for baseline glucose status? Perhaps, the use of mixed models could help here.

- Was implementation indicators measured and assessed? How about compliance to sessions? or fidelity to the intervention?

Discussion:

- Line 236-237: this sentence is not complete.

- Was the time enough to see changes in glycemia? Please refer to the DaQing Study... this need to be discussed in the paper.

- Line 262-263: What was the sample size of the mentioned studies? And the time of follow-up? And lost to follow-up rates?

-Line 272: if participants in the control group received a mini-intervention; that is not the standard of care. This need to be explained from the methods section.

- Line 273-275: Was physical activity assessed? Can you verify if the PA levels were the same at the end of the trial?

- Line 279-280: regression to the mean can be verified using the formula of Barnett AG: IJE 2004;34(1):215-20. Please, do it.

- Line 290-291: there is not clear information about what was the planned time of follow-up in the trial? Please check that.

Minor comments:

- Line 187: correct the typo: should be 246 and not 2460...

- Line 188: What does “as every second” mean in this context?

- Line 233: “tried” instead of “tryed”

- Line 289: “we” instead of “e”

Author Response

Thank you very much for your valuable comments that helped us to improve our manuscripts. We highly appreciate your comments and suggestions.

Major comments:

ABSTRACT

Comment#1: Good to clarify the duration of the intervention in some part... was that enough for glycemia reversion?

Response#1: We have now added information in regard the duration of the intervention to the abstract. Furthermore, whether the duration of the intervention is sufficient enough has been addressed in the discussion section.  Following sentence has been added to the abstract:

Line 32: “The duration of the intervention was 24 months.”

Comment#2: Prefer to use RR instead of OR to report results.

Response#2: According to the suggestions of the reviewers, we now present the RR instead of the OR throughout the manuscript.

INTRODUCTION

Comment#3: Line 45-49: Clarify the duration of the mentioned interventions. Clarify the range of potential effect reported in line 48 (I suppose 48% is the maximum effect).

Response#3: We have now added information in regard the duration of the interventions of the previous studies. We also clarified the estimation of the potential effect. Following information was added:

Line 54-57: These studies have revealed that nutritional and physical activity interventions can decrease the relative risk of T2D between 30-60% in people with IGT in controlled settings [3-9]. The length of the interventions of previously mentioned trials varied between 2.5 and 6 years [3-9].

Comment#4: Line 54-55: Add references according to the studies reported.

Response#4: The references have now been added.

Comment#5: Line 61-62: Please explain why the interventions are not expected to work well in hot and humid climate?

Response#5: We have added the following sentence to hypothesis in regard the previous statement:

Line 70-72:  “The extreme weather conditions seen in the Caribbean region (thunder storms and humid conditions) may need special logistics and motivation to improve daily physical of the study population.”

MATERIAL AND METHODS

Comment#6: Please explain what were the inclusion and exclusion criteria for the study? This part needs to coincide with the report in clinicaltrials.gov (if not, please explain why?)

Response#6: We have now revised the description of the inclusion and exclusion criteria to match correctly with the description of what we registered at clinicaltrials.gov. The text reads now as follows:

Line 112-114: “Participants 34-69 years-of-age either FINDRISC > 13 points and with a 2-hour post-challenge glucose 140-200 mg/dl (IGT); or FINDRISC > 13 points or or a fasting plasma glucose between ≥100 but <126 mg/dl (impaired fasting glucose) were included in the study.”

We also added the following sentences to clarify the difference between the inclusion criteria on clinicaltrials.gov and our analysis:

Line 114-119: “In addition to the initially defined inclusion criteria when registering the trial, we also accepted participants with isolated impaired fasting glucose (n=109) to the study. The reasons for these changes were several. First, people with IFG have a similar risk to progress to diabetes than that in people with IGT. Secondly, there is a lack of knowledge regarding the prevention potential in people with IFG, since almost all diabetes prevention trials have used IGT as the entry criterion.”

Comment#7: Give more information about the design of the study (what kind of trial was?) and about randomization (who did the randomization?  Was a third party? Or some from the team?). Who did the assignment?

Response#7: We added the required information. Following sentences were added, respectively revised:

Line 80-81: “The design of the DEMOJUAN study was a three-arm, parallel-group randomized controlled field trial using individual random allocation and was carried out in Juan Mina and Barranquilla, Northern Colombia [19].”

Line 124-128: Eligible participants were randomly allocated into one of three groups (A, B or C) with 1:1:1 allocation. The sequences for the random allocation groups were generated by IBM SPSS statistics version 19.0 for Windows. The randomization list was prepared by an independent statistician. Participants and study personnel were aware of the study-group assignments, but outcome adjudicators were not.”

Comment#8: Line 114-116 and line 121-122: some goals are reported there. Were they assessed? Were they considered as possible secondary outcomes? This can help to understand why the intervention did not work.

Response#8: Unfortunately, we were not able to assess whether the participants achieved the individual goals of the intervention as the only information we were able to obtain at the end of the intervention was the oral glucose tolerance test due to financial issues. During the intervention phase of the project we assessed participation rate in the group and individual sessions of the participants to obtain some information assessing fidelity to the interventions. We have added a new table 4 with the participation rates for the different forms of information. We have also commented upon in text (please, see response#23).

Comment#9: Line 123-125: were the participants from arm B and C combined for the seminars? How the seminars were done? What were the topics? What was the duration of the seminars? Some examples can help to understand that. In addition, a figure could help to understand what happen in each arm on the intervention.

Response#9: We have added the description of the topics of the intervention as supplementary file Appendix B to the manuscript to provide more information on the lifestyle interventions. In addition we have added following sentence:

Line 151-152: However, the seminars were held independently for the participants of group B and C.

Comment#10: Line 138-140: the paper will benefit if the questionnaire can be added to the manuscript as online supplement or something similar. Although the questionnaires were validated, questions can be relevant to understand why the intervention did not work.

Response#10: We have now added the original questionnaire as supplementary file to the manuscript (Appendix A). The questionnaire is in Spanish language and 13 pages long.

Comment#11: Line 161-164: Study outcomes: This part needs more information: Was any new OGTT done? When? Every 6 months? Or only at the end? What were the secondary outcomes if any (please check clinicaltrials.gov)? How many times was OGTT measured over time? Only at baseline and at the end? Every six months? These topics need to be clarified!

Response#11: The OGTT was measured only at the beginning and at the end (24 months) of the intervention. We have corrected that error in the manuscript.

We have added the following sentence:

Line 149: ”The OGTT was done at month 1 and month 24”

Comment#12: Secondary outcome according to clinicaltrials.gov is not considered in the analyses: "Difference in reduction of 10 year-estimated cardiovascular risk score of 10% between the standard therapy control group and the lifestyle intervention groups (WHO CVD prediction chart)". Why not to include it in the manuscript?

Response#12: Unfortunately, we were not able to obtain data of our study participants to assess the secondary outcomes project funding was scarce at the end of our project. Thus, we prioritized funding to obtain the necessary information to assess the primary outcome instead.

Comment#13: Line 166: How was the 20% difference estimate chosen? Is that estimate relevant?

Response#13: Recent scientific evidence has shown that the conversion rate from IGT to normoglycemia is as high as 50%. Studies have shown that the as compared with placebo, medication intervention might increase the possibility of conversion rate up from 50% to 70% in people with IGT. Thus, we aimed at detecting a similar difference with lifestyle interventions

We have added the following information and references now to clarify this:

199-201: “This estimate was chosen due to previous scientific evidence revealing that as compared with placebo, pharmaceutical intervention might increase the conversion rate to normoglycemia up from 50% to 70% in people with IGT [7, 41-43].”

References added:

Torgerson, J.S.; Hauptman, J., Boldrin M.N.; Sjöström, L. XENical in the Prevention of Diabetes in Obese Subjects (XENDOS) study: a randomized study of orlistat as an adjunct to lifestyle changes for the prevention of type 2 diabetes in obese patients. Diabetes Care 2004, 27(1), 155–161.

Knowler, W.C.; Hamman, R.F.; Edelstein S.L., et al. Prevention of type 2 diabetes with troglitazone in the Diabetes Prevention Program. Diabetes 2005, 54(4), 1150–1156.

Gerstein, H.C.; Yusuf, S.; Bosch J., et al. Effect of rosiglitazone on the frequency of diabetes in patients with impaired glucose tolerance or impaired fasting glucose: a randomised controlled trial. The Lancet 2006, 368(9541), 1096–1105.

Comment#14: Line 167: Is the comparison between groups? Or is it between intervention groups compared to the control one? Was the intent-to-treat principle followed in the analysis?

Response#14:  We have clarified this by adding additional information explaining the comparison between group and intention-to-treat analysis used in the study by adding the following sentences:

Line 209-210: Relative risks and their corresponding 95% confidence intervals for the intervention groups compared with the control groups were calculated for reversal to normoglycemia and T2D incidence.”

Line 205-206: The data was analyzed using SPSS statistics version 19.0 for Windows with the use of the intention-to-treat approach for all randomly assigned participants.”

Comment#15: Line 175: Why OR are calculated instead of RR? I think the manuscript will benefit to have the correct estimate here. Similarly, why the ANOVA was used instead of mixed models? In this case is required for the need of adjust estimates for difference in glucose status at baseline. Why not the secondary outcomes were assessed according to the clinicaltrials.gov?

Response#15: According to the suggestions of the reviewers, we now present the RR instead of the OR throughout the manuscript. In addition, we have analyzed the data using mixed models ANOVA. The changes in the results are marked in yellow in Table 4. We have also updated the description of the results according to new Table 4. All changes are marked in yellow.

In addition, we have added following text to the statistical methods section:

Line 212-213: “Differences in FPG and 2hPG among the three groups were assessed using mixed-design analysis of variance models in order to adjust the estimates for the difference in glucose status at baseline repeated measures.

As mentioned before, unfortunately, we were not able to obtain data of our study participants to assess the secondary outcomes project funding was scarce at the end of our project. Thus, we prioritized funding to obtain the necessary information to assess the primary outcome instead.

RESULTS

Comment#16: Relative risk should be the estimate to be reported (instead of OR).

Response#16: According to the suggestions of the reviewers, we now present the RR instead of the OR throughout the manuscript.

Comment#17: The interventions should have a positive effect on the outcome, i.e. increase the probability to revert to normoglycemia, and reduce the possibility to develop T2DM... was the codification well done in analyses?

Response#17: We carefully checked the data and codifications. We did not find any issues in that respect. Basically, the interventions per se improved the reversion to normoglycemia and reduced the conversion to T2D. However, a similar tendency was observed in the control group as well. We would like to point out that the main “problem” was that the control group improved their glucose profile without any active intervention. Therefore, no differences in the changes of the fasting and 2-hour glucose levels were observed between the intervention and control groups.

Comment#18: There is no clear information about the use of repeated measures or the use of intent-to-treat principle.

Response#18: The data was analyzed with the use of the intention-to-treat approach for all randomly assigned participant. As mentioned above, we have analyzed the data again using mixed models ANOVA adjusting for baseline glucose status according to the reviewer’s suggestions.

We have added following text to the manuscript:

Line 209-210: The data was analyzed using SPSS statistics version 19.0 for Windows with the use of the intention-to-treat approach for all randomly assigned participants.”

Line 205-206: “Differences in FPG and 2hPG among the three groups were assessed using mixed-design analysis of variance models in order to adjust the estimates for the difference in glucose status at baseline repeated measures.

Comment#19: Where are the secondary outcomes (please use info of clinicaltrials.gov, all need to be reported).

Response#19: Unfortunately, we were not able to obtain data of our study participants to assess the secondary outcomes project funding was scarce at the end of our project. Thus, we prioritized funding to obtain the necessary information to assess the primary outcome instead.

Comment#20: Figure 1: More than 50% of lost to follow-up; at the end, the study was underpowered to find the proposed differences. How affect results?

Response#20: We agree with the reviewer that the study was most likely underpowered as we lost more than 30% of the study participants. We have now added the following sentence in the limitations of the study section:

Line 375-377: “Finally, as we lost close to 50% of the study participants during the follow-up, our study may not have sufficient power to show a difference in reversion to normoglycemia or development of T2D.”

Comment#21: Table 1: what that the best way to present the age? Why not as numerical variable (i.e. mean and SD)?

Response#21: We have re-analyzed the data and present now the variable age as a continuous variable presenting mean and SD. The changes are marked in Table 1 and 2.

Comment#22: Table 4: Were these estimates adjusted for baseline glucose status? Perhaps, the use of mixed models could help here.

Response#22: Yes, according to your suggestions, we re-analyzed the data of Table 4 using mixed-design analysis of variance models in order to adjust the estimates for the difference in glucose status at baseline

Following information was added:

Line 205-206: “Differences in FPG and 2hPG among the three groups were assessed using mixed-design analysis of variance models in order to adjust the estimates for the difference in glucose status at baseline repeated measures.

Comment#23: Was implementation indicators measured and assessed? How about compliance to sessions? or fidelity to the intervention?

Response#23: During the intervention phase of the project we assessed participation rate in the group and individual sessions of the participants to obtain some information assessing fidelity to the interventions. We have added a new table 4 with the participation rates for the different forms of information.

Line 265-270: “The participation rates in the interventions sessions varied according to the type of the session. In general, three out of four intervention group participants took part in at least one group and one individual intervention session. Whereas 32% of the nutritional intervention group participants came to at least 50% of all planned individual sessions, only 9% had an at least 50% participation rate for the group sessions. The corresponding participation rates for the physical activity study participants were 46% (individual sessions) and 15% (group sessions).”

Table 4. Participation rates of the study participants in the interventions

Group

Control

Nutrition

Physical activity

(n=122)

(n=136)

(n=132)

% (n)

% (n)

% (n)

Participated in at least in one group and one   individual intervention session

0 (0)

74 (100)

76 (100)

Participated in at least 50% of all individual sessions   sessions

0 (0)

32 (43)

46 (61)

Participated in at least 50% of all group sessions

0 (0)

9 (12)

15 (20)

In addition, we have address this issue in a small paragraph in the discussion section:

Line 332-337: “Furthermore, compared with randomized controlled trials, it is very difficult to reach a very high compliance (or participation rates) to the interventions in field trials as the study population cannot be controlled in a similar way. However, our study showed that interventions can be implemented within the primary-health care system and that at least reaching acceptable compliance rates for individual sessions may be achieved. Naturally, the reasons for non-compliance needs to be assessed further in order to adapt the program.

DISCUSSION

Comment#23: Line 236-237: this sentence is not complete.

Response #23: We have completed the sentence that states now as follows:

Line 257-259: Similarly, no statistically significant association was found between the intervention groups and the incidence of type 2 diabetes.”

Comment#24: Was the time enough to see changes in glycemia? Please refer to the DaQing Study... this need to be discussed in the paper.

Response#24: We agree that length of intervention as well as the study design may explain partly our findings. We have added the following information to the discussion to elaborate more on that issue:

Line 326-339: One important factor explaining the differences of the results between randomized controlled clinical trials and their implementation in the general population is length of intervention. Whereas the length of the interventions of the randomized clinical trials varied between 2.5 and 6 years [3-9], the duration of the interventions of the pragmatic or field trials in the “real-world” were of shorter duration [10-18]. Implementing randomized controlled clinical trials in the population setting have shown to have less impact, among others due to a higher heterogeneity of the study population and compliance to the interventions. Furthermore, compared with randomized controlled trials, it is very difficult to reach a very high compliance (or participation rates) to the interventions in field trials as the study population cannot be controlled in a similar way. However, our study showed that interventions can be implemented within the primary-health care system and that at least reaching acceptable compliance rates for individual sessions may be achieved. Naturally, the reasons for non-compliance needs to be assessed further in order to adapt the program. Thus, this and our short length of the intervention may be some additional reasons why we were not able to show any differences in changes of the glucose levels between the intervention and the control group.”

Comment#25: Line 262-263: What was the sample size of the mentioned studies? And the time of follow-up? And lost to follow-up rates?

Response#25: We have added the requested information to the text. The text states now the following:

Line 297-299:  These changes were maintained through 25-32 months of follow-up [10]. The initially included 92 study participants and their documented loss to follow-up during the first 12 months was 37% for the intervention group and 28% for the control group.

Comment#26: Line 272: if participants in the control group received a mini-intervention; that is not the standard of care. This need to be explained from the methods section.

Response#26: We have added the suggested information to the methods section. Following sentence has been added:

Line 130-132: In addition, the participants in the control group were told about their high risk of T2D and received “mini-intervention” to improve their lifestyle.

Comment#27: Line 273-275: Was physical activity assessed? Can you verify if the PA levels were the same at the end of the trial?

Response#27: Unfortunately, we were not able to measure the physical activity levels or any other cardio-metabolic variable of the study participants at the end of the trial due to lack of funding. There was an unforeseen increase in the exchange rate between the USD and the Colombian peso that heavily affected our study. Thus, we were prioritizing to assess the primary outcome of our study (glucose levels).

Comment#28: Line 279-280: regression to the mean can be verified using the formula of Barnett AG: IJE 2004;34(1):215-20. Please, do it.

Response#28: Actually, we realized the RTM may not be a possible reason as study participants were randomly allocated to the intervention and comparison groups the impact of the intervention would be equally affected by regression to the mean as indicated by the article you mentioned. We have added the following sentence referring to Barnett et al:

Line 358-360: “However, as study participants were randomly allocated to the intervention and comparison groups the impact of the intervention would be equally affected by regression to the mean [45].

Comment#29: Line 290-291: there is not clear information about what was the planned time of follow-up in the trial? Please check that.

Response#29: Their planned follow-up of the trial was 12 months. We have now added that information. The sentence reads now as follows:

Line 325: Unlike in our study, they reported reductions in FPG and 2-hPG in their study participants during their 12-month intervention.

MINOR COMMENTS

Comment#30: Line 187: correct the typo: should be 246 and not 2460...

Response#30: We have corrected the typo.

Comment#31: Line 188: What does “as every second” mean in this context?

Response#31: We have revised the sentence as follows: “Half of the study participants…”

Comment#32: Line 233: “tried” instead of “tryed”

Response#32: We have corrected the typo.

Comment#33: Line 289: “we” instead of “e”

Response#33: We have corrected the typo.

Reviewer 2 Report

I suggest the author in line 111 should explain the lifestyle intervention.

Line 114-117 I suggest the author should provide a brief explanation of how these goals mentioned were monitored and evaluated.

How was the Physical activity intervention monitored and evaluated on a day to day basis for the period of the study

What was the attrition rate for the study in each group?

How reliable and valid were the anthropometric and blood pressure measurements

Table 1- What was the rationale for dividing the groups by age group? I suggest they be divided by gender given the dynamic of metabolic function of each gender.

What was the age range in each group particularly less than 45 years and over 64 years?

Line 186- Random allocation of groups were made. I suggest the author should briefly explain this process and indicate if it was single or double blinded.

Line 187 correct the typing error of the number of subjects for the control

Line 232- correct the typographical error at the end of the sentence

One wonders if members of the three groups were from the same community in some instances.

The manuscript needs language editing

Author Response

Thank you very much for your valuable comments that helped us to improve our manuscripts. We highly appreciate your comments and suggestions.

Comment#1: I suggest the author in line 111 should explain the lifestyle intervention.

Response#1: In agreement with both reviewers, we have added a more detailed description of the intervention as supplementary file Appendix B not to extend the length of the manuscript too much.

Comment#2: Line 114-117 I suggest the author should provide a brief explanation of how these goals mentioned were monitored and evaluated.

Response#2: These lines refer to the randomization process of our study. We have now clarified this by revising the information. The new, revised text states now as follows:

Line 124-128: “Eligible participants were randomly allocated into one of three groups (A, B or C) with 1:1:1 allocation. The sequences for the random allocation groups were generated by IBM SPSS statistics version 19.0 for Windows. The randomization list was prepared by an independent statistician. Participants and study personnel were aware of the study-group assignments, but outcome adjudicators were not.”

In addition, the issue in regard monitoring cardio metabolic indicators during the intervention is addressed below.

Comment#3: How was the Physical activity intervention monitored and evaluated on a day to day basis for the period of the study

Response#3: Unfortunately, we were not able to measure the physical activity levels or any other cardio-metabolic variable of the study participants during or at the end of the trial due to lack of funding. There was an unforeseen increase in the exchange rate between the USD and the Colombian peso that heavily affected our study. Thus, we were prioritizing to assess the primary outcome of our study (glucose levels). This may reflect the usual problems in implementing research in countries of economic transition.

Comment#4: What was the attrition rate for the study in each group?

Response#4: The participation rates in the interventions sessions varied according to the type of the session. In general, three out of four intervention group participants took part in at least one group and one individual intervention session. Whereas 32% of the nutritional intervention group participants came to at least 50% of all planned individual sessions, only 9% had an at least 50% participation rate for the group sessions. The corresponding participation rates for the physical activity study participants were 46% (individual sessions) and 15% (group sessions).

Table 4. Participation rates of the study participants in the interventions

Group

Control

Nutrition

Physical activity

(n=122)

(n=136)

(n=132)

% (n)

% (n)

% (n)

Participated in at least in   one group and one individual intervention session

0 (0)

74 (100)

76 (100)

Participated in at least 50%   of all individual sessions sessions

0 (0)

32 (43)

46 (61)

Participated in at least 50%   of all group sessions

0 (0)

9 (12)

15 (20)

In addition, we have address this issue in a small paragraph in the discussion section:

Line 332-337: “Furthermore, compared with randomized controlled trials, it is very difficult to reach a very high compliance (or participation rates) to the interventions in field trials as the study population cannot be controlled in a similar way. However, our study showed that interventions can be implemented within the primary-health care system and that at least reaching acceptable compliance rates for individual sessions may be achieved. Naturally, the reasons for non-compliance needs to be assessed further in order to adapt the program.

Comment#5: How reliable and valid were the anthropometric and blood pressure measurements

Response#5: We believe that the anthropometric and blood pressure measurements were valid and within the usual range of measurement errors in this type of research. All measurements used standardized protocols and the staff taking the measurements were professionals with long-standing experience in measuring anthropometry and blood pressure.

We have added following sentence:

Line 170-171: “All measurements followed a standardized protocol and the staff taking the measurements were professionals with long-standing experience in measuring anthropometry and blood pressure.

Comment#6: Table 1- What was the rationale for dividing the groups by age group? I suggest they be divided by gender given the dynamic of metabolic function of each gender.

Response#6: We have re-analyzed the data and present now the variable age as a continuous variable presenting mean and SD to comply with the comments of both reviewer#1 and #2. The changes are marked in Table 1 and 2.

Comment#7: What was the age range in each group particularly less than 45 years and over 64 years?

Response#7: We have re-analyzed the data and present now the variable age as a continuous variable presenting mean and SD to comply with the comments of both reviewer#1 and #2. The changes are marked in Table 1 and 2.

Comment#8: Line 186- Random allocation of groups were made. I suggest the author should briefly explain this process and indicate if it was single or double blinded.

Response#8: We have added the following information to clarify the randomization and blinding process.

Line 124-128: Eligible participants were randomly allocated into one of three groups (A, B or C) with 1:1:1 allocation. The sequences for the random allocation groups were generated by IBM SPSS statistics version 19.0 for Windows. The randomization list was prepared by an independent statistician. Participants and study personnel were aware of the study-group assignments, but outcome adjudicators were not.”

Comment#9: Line 187 correct the typing error of the number of subjects for the control

Response#9: We have corrected the typo.

Comment#10: Line 232- correct the typographical error at the end of the sentence

Response#10: We have corrected the typo.

Comment#11: One wonders if members of the three groups were from the same community in some instances.

Response#11: This is an excellent observation and indeed they were, this may be one of the reasons why the control group improved their glucose profile as well. In order to address this, we have added the following sentence to the discussion section:

Line 348-350: Moreover, in some instance members of the three groups were from the same community, thus a dilution of the intervention from the intervention groups to the control group participants may have occurred.

Comment#12: The manuscript needs language editing

Response#12: The language has been edited throughout the manuscript.

Reviewer 3 Report

Please see my comments below:

1) How did the authors define “the people at high risk of diabetes”?

2) Line 49: please explain the importance of having studies included Latin America populations.

3) Line 91: Please explain the scores values of the total risk. Do the scores of 226 mean high risk?

4) Did the authors calculate the minimum sample size needed?

5) Please consider the intention-to-treat analysis in the study.

6) Lines 184, 233 and 289: please correct the typo error.

7) Discussion: the authors should discuss why their primary findings were insignificant in detail. The current explanation is a bit vague.

8) Did the authors explain about the strengths of their study?

9) The structure of the discussion did not flow well. Please reorganize them. It is confusing for the readers to follow the discussion.

10) please summarise the reasons provided by the dropouts.

Author Response

Thank you very much for your valuable comments that helped us to improve our manuscripts. We highly appreciate your comments and suggestions.

Comment#1: How did the authors define “the people at high risk of diabetes”?

Response#1: We defined high risk of type 2 diabetes as having impaired glucose tolerance. We used the FINDRISC to identify these people in a two-step screening procedure (FINDRISC = confirmatory blood tests). In order to be more specific and comply with your observation, we have changed the title of our manuscript to:

Early lifestyle interventions in people with impaired glucose tolerance in Northern Colombia: The DEMOJUAN project.

Comment#2: Line 49: please explain the importance of having studies included Latin America populations.

Response#2: We have now added some more information on the importance of having studies included Latin America populations in the introduction in the two locations of the introduction:

Line 48-50: “According to recent data in 2017, the national prevalence of type 2 diabetes (T2D) in Colombia was 8.1% for the 20–79 year-old population imposing a heavy financial burden for the health-care system [1].”

Line 57: “However, none of these studies included Latin American populations.”

Line 65-72: “While previous findings offer a compelling evidence-base, it is necessary to learn as to how the prevention of T2D works in real life in different settings and population groups. In addition, it is important to find out how well the lifestyle interventions works within the primary health-care system in Latin America. Thus far, such trials have not been carried out in the Caribbean region of Latin America and it is not known whether these lifestyle interventions will work well in a hot and humid climate, in particular the physical activity component. The extreme weather conditions seen in the Caribbean region (thunder storms and humid conditions) may need special logistics and motivation to improve daily physical of the study population.

Comment#3: Line 91: Please explain the scores values of the total risk. Do the scores of 26 mean high risk?

Response#3: Previous studies have shown that a FINDRISC of > 13 points corresponds to a high risk of having IGT/IFG or diabetes. We have added following clarification:

Line 103-104: “A FINDRISC of > 13 points was considered as high risk of having IGT, IFG or diabetes.”

Comment#4: Did the authors calculate the minimum sample size needed?

Response#4: Yes, we did. The sample size calculations are presented in the first paragraph of the statistical analysis section that states:

Line 197-204:  The study was designed to have a 90% power to detect a 20% percentage unit difference in recovery from IGT (comparing 70% vs. 50% recovery rates) between the treatment groups at 5% significance level. This estimate was chosen due to previous scientific evidence revealing that as compared with placebo, pharmaceutical intervention might increase the conversion rate to normoglycemia up from 50% to 70% in people with IGT [7, 41-43]. Assuming a 30% loss to follow up at the end of the 24 months intervention, a total of 200 participants are needed in both treatment groups (total sample size of 600 individuals). The estimated drop-out of 30% was decided according to the results and experiences of previous randomized controlled diabetes trials in Europe and Unites States [6, 11, 44].”

Comment#5: Please consider the intention-to-treat analysis in the study.

Response#5: We have clarified this by adding additional information explaining the comparison between group and intention-to-treat analysis used in the study by adding the following sentences:

Line 205: The data was analyzed using SPSS statistics version 19.0 for Windows with the use of the intention-to-treat approach for all randomly assigned participants.”

Comment#6: Lines 184, 233 and 289: please correct the typo error.

Response#6: The typos have been corrected.

Comment#7: Discussion: the authors should discuss why their primary findings were insignificant in detail. The current explanation is a bit vague.

Response#7: We have now elaborated a bit more on that item. Following paragraph has now been added:

Line 326-339: “One important factor explaining the differences of the results between randomized controlled clinical trials and their implementation in the general population is length of intervention. Whereas the length of the interventions of the randomized clinical trials varied between 2.5 and 6 years [3-9], the duration of the interventions of the pragmatic or field trials in the “real-world” were of shorter duration [10-18]. Implementing randomized controlled clinical trials in the population setting have shown to have less impact, among others due to a higher heterogeneity of the study population and compliance to the interventions. Furthermore, compared with randomized controlled trials, it is very difficult to reach a very high compliance (or participation rates) to the interventions in field trials as the study population cannot be controlled in a similar way. However, our study showed that interventions can be implemented within the primary-health care system and that at least reaching acceptable compliance rates for individual sessions may be achieved. Naturally, the reasons for non-compliance needs to be assessed further in order to adapt the program. Thus, this and our short length of the intervention may be some additional reasons why we were not able to show any differences in changes of the glucose levels between the intervention and the control group.”

Comment#8: Did the authors explain about the strengths of their study?

Response#8: We have added now a small paragraph about the strengths of our study right before the description of the limitation section. The paragraph states now as follows:

Line xx: “One of the strengths of our study was that we managed to design and implement a field trial within a primary health-care system of a country of economic transition. Close to 50% of the people with prediabetes at the beginning of the study had their glucose values normalized at the end of the study. This shows that early lifestyle intervention programs may prevent or delay the development of T2D and can be successfully integrated into the primary health care system.”

Comment#9: The structure of the discussion did not flow well. Please reorganize them. It is confusing for the readers to follow the discussion.

Response#9: We have now revised the discussion according to all three reviewers’ comments as much as we could to make the reading easier and adding the additional information asked by the reviewers.

Comment#10: Please summarize the reasons provided by the dropouts.

Response#10: Unfortunately, we do not have any information on the drop-outs. We assume that some of them moved somewhere else or just lost interest in participating. However, depending on additional funding, we may try to contact some of them to conduct a qualitative study assessing reasons for the drop-out.

Round 2

Reviewer 1 Report

No comments.